# GENERATING REALISTIC STOCK MARKET ORDER STREAMS

## ABSTRACT

We propose an approach to generate realistic and high-fidelity stock market data based on generative adversarial networks. We model the order stream as a stochastic process with finite history dependence, and employ a conditional Wasserstein GAN to capture history dependence of orders in a stock market. We test our approach with actual market and synthetic data on a number of different statistics, and find the generated data to be close to real data.

## 1 INTRODUCTION

Financial markets are among the most well-studied and closely watched complex multiagent systems in existence. Well-functioning financial markets are critical to the operation of a complex global economy, and small changes in the efficiency or stability of such markets can have enormous ramifications. Accurate modeling of financial markets can support improved design and regulation of these critical institutions. There is a vast literature on financial market modeling, though still a large gap between the state-of-art and the ideal. Analytic approaches provide insight through highly stylized model forms. Agent-based models accommodate greater dynamic complexity, and are often able to reproduce "stylized facts" of real-world markets (LeBaron, 2006). Currently lacking, however, is a simulation capable of producing market data at high fidelity and high realism. Our aim is to develop such a model, to support a range of market design and analysis problems. This work provides a first step, learning a high-fidelity generator from real stock market data streams.

Our main contribution is an approach to produce stock market data that is close to real market data, using a Wasserstein generative adversarial network (WGAN) (Arjovsky et al., 2017). There are many challenges that we overcome as part of this contribution. The first is how to represent a stream of stock market orders as data that can be used in a WGAN. Towards this end, we assume the stock market data stream to arise from a stochastic process with finite (but long) memory dependence. The stochastic process view also makes precise the conditional distribution that the generator is learning as well the joint distribution that the critic of the WGAN distinguishes by estimating the earth-mover distance.

The second main challenge is the design of the network architecture. We choose a conditional WGAN to capture the history dependence of the stochastic process, with both the generator and critic conditional on history of orders and the time of day. A single LSTM layer is used to summarize the history succinctly. The internal architecture of both the generator and critic uses a standard convolutional structure. The generator outputs the next stock market order as well as how this order changes the active orders in the market. Part of the generator output, which updates the active market orders, is produced using a pre-trained network to approximate the deterministic buy and sell order matching in the stock market.

Finally, we experiment with synthetic and real market data. The synthetic data is produced using a stock market simulator that has been used in several agent-based financial studies. The real data was obtained from OneMarketData, a financial data provider and publisher of the OneTick database product. We evaluate the generated data using various statistics such as the distribution of price and quantity of orders, inter-arrival times of orders, and the best bid and best ask evolution over time. We find the generated data matches the corresponding statistics in real data (simulated or actual stock market) closely.

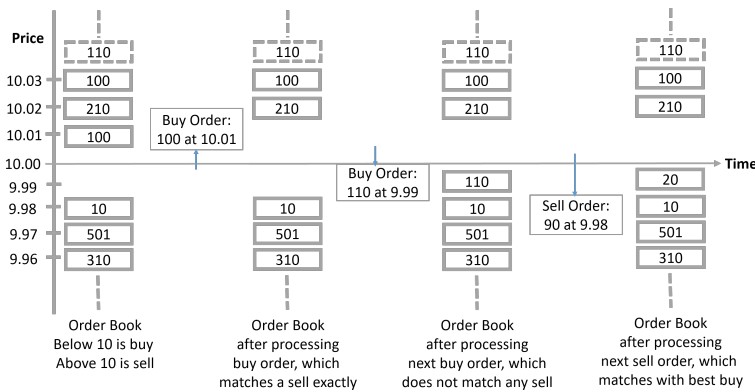

Figure 1: Visual representation and evolution of a limit order book.

## 2 RELATED WORK AND BACKGROUND

WGAN is a popular and well-known variant of GANs (Goodfellow et al., 2014). Most prior work on generation of sequences using GANs has been in the domain of text generation (Press et al., 2017; Zhang et al., 2017). However, since the space of word representations is not continuous, the semantics change with nearby word representation, and given a lack of agreement on the metrics for measuring goodness of sentences, producing good quality text using GANs is still an active area of research. Stock market data does not suffer from this representation problem but the history dependence for stock markets can be much longer than for text generation. In a sequence of recent papers, Xiao et al. (2017; 2018) have introduced GAN-based methods for generating point processes. The proposed methods generate the time for when the next event will occur. The authors have also explored the use of these techniques to generate the time for transaction events in stock markets. Our problem is richer as we aim to generate the actual limit orders including time, order type, price, and quantity information.

Deep neural networks and machine learning techniques have been used on financial data mostly for prediction of transaction price (Hiransha et al., 2018; Bao et al., 2017; Qian, 2017) and for prediction of actual returns (Abe & Nakayama, 2018). As stated, our goal is not market prediction per se, but rather market modeling. Whereas the problems of learning to predict and generate may overlap (e.g., both aim to capture regularity in the domain), the evaluation criteria and end product are quite distinct.

The stock market is a venue where equities or stocks of publicly held companies are traded. Nearly all stock markets follow the *continuous double auction* (CDA) mechanism (Friedman, 1993). Traders submit bids, or *limit orders*, specifying the maximum price at which they would be willing to buy a specified quantity of a security, or the minimum price at which they would be willing to sell a quantity.[1] The *order book* is a store that maintains the set of active orders: those submitted but not yet transacted or canceled. CDAs are continuous in the sense that when a new order matches an existing (incumbent) order in the order book, the market clears immediately and the trade is executed at the price of the incumbent order—which is then removed from the order book. Orders may be submitted at any time, and a buy order matches and transacts with a sell order when the limits of both parties can be mutually satisfied. For example, as shown in Figure 1 if a limit buy order with price $10.01 and quantity 100 arrives and the order book has the best offered sell price at $10.01 with quantity 100 then the arriving order matches an incumbent exactly. However, the next buy order does not match any sell, and the following sell order partially matches what is then the best buy in the order book.

The limit order book maintains the current active orders in the market (or the state of the market), which can be described in terms of the quantity offered to buy or sell across the range of price levels. Each order arrival changes the market state, recorded as an update to the order book. After processing any arrived order every buy price level is higher than all sell price levels, and the *best bid*

---

[1]Hence, the CDA is often referred to as a limit-order market in the finance literature (Abergel et al., 2016).

refers to the lowest buy price level and the *best ask* refers to the highest sell price level. See Figure 1 for an illustration. The order book is often approximated by few (e.g., ten) price levels above the best bid and ten price levels below the best ask; as these prices are typically the ones that dictate the transactions in the market. There are various kinds of traders in a stock market, ranging from individual investors to large investing firms. Thus, there is a wide variation in the nature of orders submitted for a security. We aim to generate orders for a security in aggregate (not per agent) that is close to the aggregate orders generated in a real market. We focus on generating orders and do not attempt to generate transactions in the stock market. This is justified as the CDA mechanism is deterministic and transactions can be determined exactly given a stream of orders.

## 3 STOCK-GAN

### 3.1 STOCK MARKET ORDERS AS A STOCHASTIC PROCESS

We model stock market orders as a stochastic process. Recall that a stochastic process is a collection of random variables indexed by a set of numbers. We view the stock market orders for a given chunk of time of day $\Delta t$ as a collection of vector valued random variable $\{\mathbf{x}_i\}_{i \in N}$ indexed by the limit order sequence number in $N = \{1, \ldots, n\}$. The components of the random vector $\mathbf{x}_i$ include the time interval $d_i$, type of order $t_i$, limit order price $p_i$, limit order quantity $q_i$, and the best bid $a_i$ and best ask $b_i$. The time interval $d_i$ specifies the difference in time between the current order $i$ and previous order $i - 1$ (in precision of milliseconds); the range of $d_i$ is finite. The type of order can be buy, sell, cancel buy, or cancel sell (represented in two bits). The price and quantity are restricted to lie within finite bounds. The price range is discretized in units of US cents and the quantity range is discretized in units of the equity (non-negative integers). The best bid and best ask are limit orders themselves and are specified by price and quantity. Observe that we assume the stochastic process depends on the discrete time of day $\Delta t$, which we will make explicit in the next paragraph. We divide the time in a day into 25 equal intervals and $\Delta t$ refers to the index of the interval. A visual representation of $\mathbf{x}_i$ is shown in Figure 2(a).

Following the terminology prevalent for stochastic processes, the above process is discrete time and discrete space (note that discrete time in this terminology here refers to the discreteness of the index set $N$). We assume a finite history dependence of the current output $\mathbf{x}_i$, that is, $P(\mathbf{x}_i \mid \mathbf{x}_{i-1}, \ldots, \Delta t) = P(\mathbf{x}_i \mid \mathbf{x}_{i-1}, \ldots, \mathbf{x}_{i-m}, \Delta t)$. Such dependence is justified by the observation that recent orders mostly determine the transactions and transaction price in the market as orders that have been in the market for long either get transacted or canceled. Further, the best bid and best ask serves as an (approximate) sufficient statistic for events beyond the history length $m$. While this process is not a Markov chain, it forms what is known as a higher order Markov chain, which implies that the process given by $\mathbf{y}_i = (\mathbf{x}_i, \ldots, \mathbf{x}_{i-m+1})$ is a Markov chain for any given time interval $\Delta t$. We assume that this chain formed by $\mathbf{y}_i$ has a stationary distribution (i.e., it is irreducible and positive recurrent). A Markov chain is a *stationary stochastic process* if it starts with its stationary distribution. After some initial mixing time, the Markov chain does reach its stationary distribution, thus, we assume that the process is stationary by throwing away some initial data for the day. Also, for the jumps across two time intervals $\Delta t$, we assume the change in stationary distribution is small and hence the mixing happens very quickly. A stationary process means that $P(\mathbf{x}_i, \ldots, \mathbf{x}_{i-m+1} \mid \Delta t)$ has the same distribution for any $i$. In practice we do not know $m$. However, we can assume a larger history length $k + 1 > m$, and then it is straightforward to check that $\mathbf{y}_t = (\mathbf{x}_i, \ldots, \mathbf{x}_{i-k})$ is a Markov chain and the claims above hold with $m - 1$ replaced by $k$. We choose $k = 20$.

### 3.2 WGAN ARCHITECTURE AND WORKING

Given the above stochastic process view of the problem, we design a conditional WGAN with a recurrent architecture to learn the real conditional distribution $P_r(\mathbf{x}_i \mid \mathbf{x}_{i-1}, \ldots, \mathbf{x}_{i-k}, \Delta t)$. We use the subscript $r$ to refer to real distributions and the subscript $g$ to refer to generated distributions. The real data $\boldsymbol{x}_1, \boldsymbol{x}_2, \ldots$ is a realization of the stochastic process. It is worth noting that even though $P(\mathbf{x}_i, \ldots, \mathbf{x}_{i-k} \mid \Delta t)$ has the same distribution for any $i$, the realized real data sequence $\boldsymbol{x}_i, \ldots \boldsymbol{x}_{i-k}$ is correlated with any overlapping sequnce $\boldsymbol{x}_{i+k'}, \ldots \boldsymbol{x}_{i-k+k'}$ for $k \geq k' \geq -k$. Our data points for training (stated in detail in the next paragraph) are sequences $\boldsymbol{x}_i, \ldots \boldsymbol{x}_{i-k}$, and to ensure independence in a batch we make sure that the sequences chosen in a batch are sufficiently far apart.

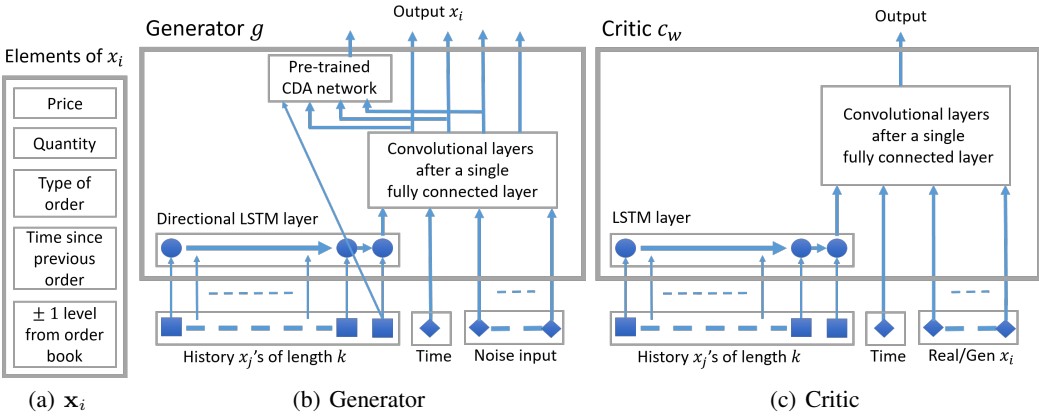

Figure 2: Stock-GAN architecture

**Architecture**: The architecture is shown in Figure 2. Our proposed WGAN is conditional (Mirza & Osindero, 2014) with both the generator and critic conditioned on a $k$ length history and the time interval $\Delta t$. The history is condensed to one vector using a single LSTM layer. This vector and some uniform noise is fed to a fully connected layer layer followed by a convolutional structure. The generator outputs the next $\boldsymbol{x}_i$ (realization of $\mathbf{x}_i$) and the critic outputs one real number. Note that when training both generator and critic are fed history from real data, but when the generator executes after training it is fed its own generated data as history. As stated earlier, the generator output includes the best bid and ask. As the best bid and ask can be inferred deterministically from the current order and the previous best bid and ask (for most orders), we use another neural network (with frozen weights during GAN training) to output the best bid and ask. We call this the CDA network. The CDA network is trained separately using a standard MSE loss (see Appendix C).

**Critic details**: When fed real data, the critic can be seen as a function $c_w$ of $\boldsymbol{s}_i = (\boldsymbol{x}_i, \ldots, \boldsymbol{x}_{i-k}, \Delta \boldsymbol{t})$, where $w$ are the weights of the critic network. As argued earlier, samples in a batch that are chosen from real data that are spaced at least $k$ apart are i.i.d. samples of $P_r$. Then for $m$ samples fed to the critic, $\frac{1}{m} \sum_{i=1}^{m} c_w(\boldsymbol{s}_i)$ estimates $E_{\mathbf{s} \sim P_r}(c_w(\mathbf{s}))$. When fed generated data (with the ten price levels determined from the output order and previous ten levels), by similar reasoning $\frac{1}{m} \sum_{i=1}^{m} c_w(\boldsymbol{s}_i)$ estimates $E_{\mathbf{s} \sim P_g}(c_w(\mathbf{s}))$ when the samples are sufficiently apart (recall that the history is always real data). Thus, the critic computes the Wasserstein distance between the joint distributions $P_r(\mathbf{x}_i, \ldots, \mathbf{x}_{i-k}, \Delta t)$ and $P_g(\mathbf{x}_i, \ldots, \mathbf{x}_{i-k}, \Delta t)$. Further, we use a gradient penalty term in the loss function for the critic instead of clipping weights as proposed in the original WGAN paper (Arjovsky et al., 2017) because of the better performance as revealed in prior work (Gulrajani et al., 2017).

**Generator details**: The generator learns the conditional distribution $P_g(\mathbf{x}_i \mid \mathbf{x}_{i-1}, \ldots, \mathbf{x}_{i-k}, \Delta t)$. Along with the real history, the generator represents the distribution $P_g(\mathbf{x}_i, \ldots, \mathbf{x}_{i-k}, \Delta t) = P_g(\mathbf{x}_i \mid \mathbf{x}_{i-1}, \ldots, \mathbf{x}_{i-k}, \Delta t) P_r(\mathbf{x}_{i-1}, \ldots, \mathbf{x}_{i-k}, \Delta t)$.

The loss functions used is the standard WGAN loss function with a gradient penalty term (Gulrajani et al., 2017). The critic is trained 100 times in each iteration and as already stated, the notable part in constructing the training data is that for each mini-batch the sequence of orders chosen (including history) is far away from any other sequence in that mini-batch (see Appendix C for code snippets).

# 4 EXPERIMENTAL RESULTS

We apply and evaluate Stock-GAN on two types of data sets composed of orders from an agent-based market simulator and from a real stock market, respectively. We describe each data set in detail and then compare key metrics and distributions of our generated orders with ground truth orders from the agent-based simulator and real stock markets.

## 4.1 SYNTHETIC AND REAL DATA

**Synthetic data**: We first evaluate Stock-GAN on synthetic orders generated from an agent-based market simulator. Previously adopted to study a variety of issues in financial markets (e.g., market making and manipulation), the simulator captures stylized facts of the complex financial market with specified stochastic processes and distributions (Wellman & Wah, 2017). We briefly describe the market simulator below.

In the simulation, the market operates over a finite time horizon. Agents enter and reenter the market according to a Poisson process with an arrival rate of 0.005. On each arrival these traders submit a limit order to the market (replacing their previous order, if any), indicating the price at which they are willing to buy or sell a single unit of the security. The market environment is populated by 32 traders, representing investors. Each investor has an individual valuation for the security made up of private and common components. The common component is represented by a fundamental value, which can be viewed as the intrinsic value of the security. This fundamental value varies over time according to a mean-reverting stochastic process. The private component of value captures the preference contribution of the individual agent's reason for trading this security at the current time (e.g., investment, liquidity, diversification). The private valuations are drawn from a specified distribution at the start of a simulation. The common and private components are effectively added together to determine each agents valuation of the security. Agents accrue private value on each transaction, and at the end of the trading horizon evaluate their accumulated inventory on the basis of a prediction of the end-time fundamental. Given the market mechanism and valuation model for the simulation, investors pursue their trading objectives by executing a trading strategy in that environment. A popular trading strategy we adopt in the simulator is the *zero-intelligence* (ZI) strategy (Farmer et al., 2005). The ZI trader shades its bid from its current valuation of the stock by a random offset. We use about 300,000 orders generated by the simulator as our synthetic data. The price output by the simulator is normalized to lie in the interval $[-1, 1]$.

**Real data**: We obtained real limit-order streams from OneMarketData, who provided access for our research to their OneTick database for selected time periods and securities. The provided data streams comprise order submissions and cancellations across multiple exchanges at millisecond granularity. In experiments, we evaluate in the performance of Stock-GAN on two securities: a small capitalization stock, Patriot National (PN), and a large capitalization stock, Alphabet Inc (GOOG). The two stocks differ in several key aspects, including investment sector, market activity intensity, price range, liquidity etc., and thus their order patterns represent distinct dynamic processes. By training Stock-GAN with historical data for individual stocks, we can generate limit-order streams that capture key characteristics of each.

Relative to our simulated agent-based market, the real market limit orders tend be very noisy including many orders at extreme prices far from the range where transactions occur. Since our interest is primarily on behavior that can affect market outcomes, we focus on limit orders in the relevant range near the best bid and ask. Specifically, in a preprocessing step, we eliminate limit orders that never appear within ten levels of the best bid and ask prices. In the experiment reported here, we use historical real market data of PN during one trading day in August 2016, and GOOG during one trading day in August 2017. After preprocessing, the PN daily order stream has about 20,000 orders and GOOG has about 230,000.

## 4.2 EVALUATION STATISTICS

We generate a number of orders equal to the number of real orders used to train the WGAN. We evaluate our generated order stream in comparison to real data using the following statistics:

1. Price. Distribution over price for the day's limit orders, by order type.
2. Quantity. Distribution over quantity for the day's limit orders, by order type.
3. Inter-arrival time. Distribution over inter-arrival durations for the day's limit orders, by order type.
4. Intensity evolution. Number of orders for consecutive 1000-second chunks of time.
5. Best bid/ask evolution. Changes in the best bid and ask over time as new orders arrive.

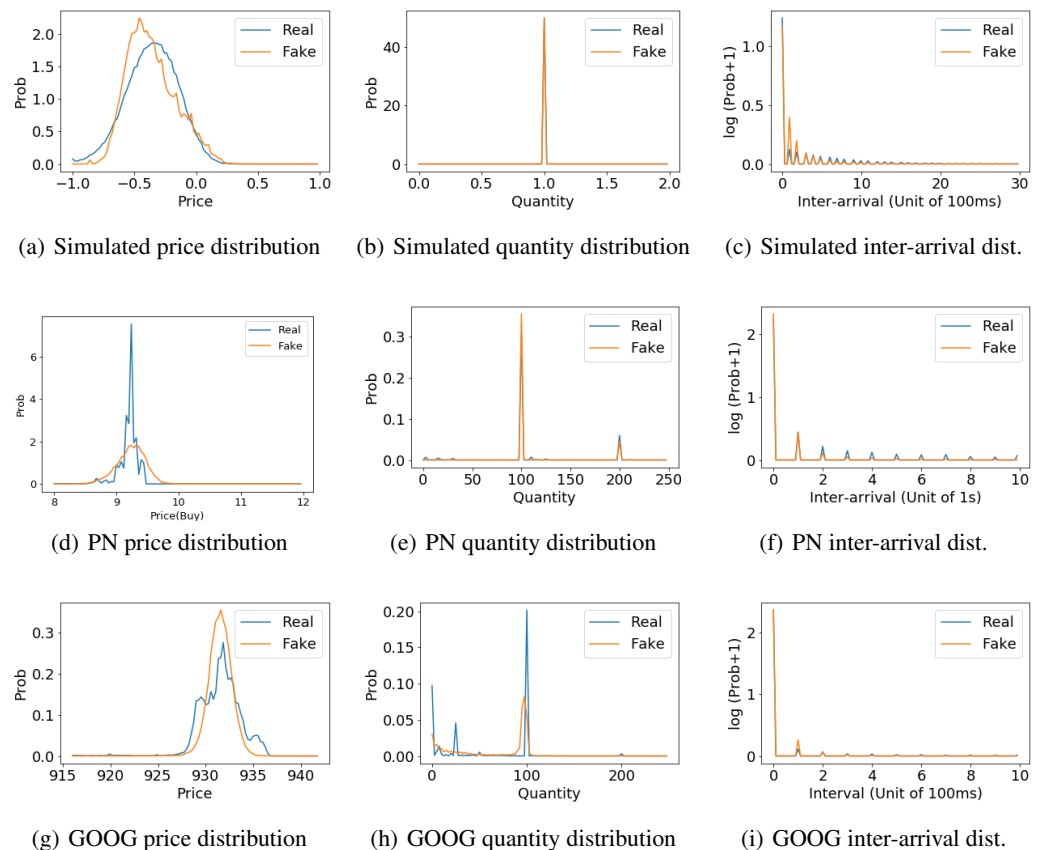

Figure 3: Simulated, PN, and GOOG submitted buy-order statistics.

**A note on cancellation:** In our generation process, cancellation type orders are not contingent on the order book. We use a heuristic which is to match the generated cancellation order to the closest priced order in the book. Cancellations that are too far from any existing order to be a plausible match are ignored.

### 4.3 RESULTS

In describing our results, "real" refers to simulated or actual stock market data and "fake" refers to generated data. Figure 3 presents statistics on buy orders for the three cases when the real data is simulated, PN, or GOOG. For simulated data, the price and inter-arrival distribution matches the real distribution quite closely. The quantity for the simulated data is always one, which is also trivially captured in the generated data. For PN and GOOG, the quantity distribution misses out on some peaks but gets most of the peaks in the real distribution. The inter-arrival time distribution matches quite closely (note that the axis has been scaled for inter-arrival time to highlight the peaks and show the full range of time). The price distribution matches closely for GOOG, but is slightly off for PN, which could be due to the low amount of data for PN.

Figure 4 presents statistics on sell orders for the three cases when the real data is simulated, PN, or GOOG. The results for sell orders are quite similar to buy orders. Results for cancellations are included in the appendix.

Figure 5 presents order intensity as a function of time (number of orders in every chunk of 1000 secs normalized by max number) for the simulated, PN, and GOOG markets. As in the graphs for other statistics, generated WGAN results are compared with the measured intensities in the real data. The intensities show similar trends, though for the real markets there is significant variation. The

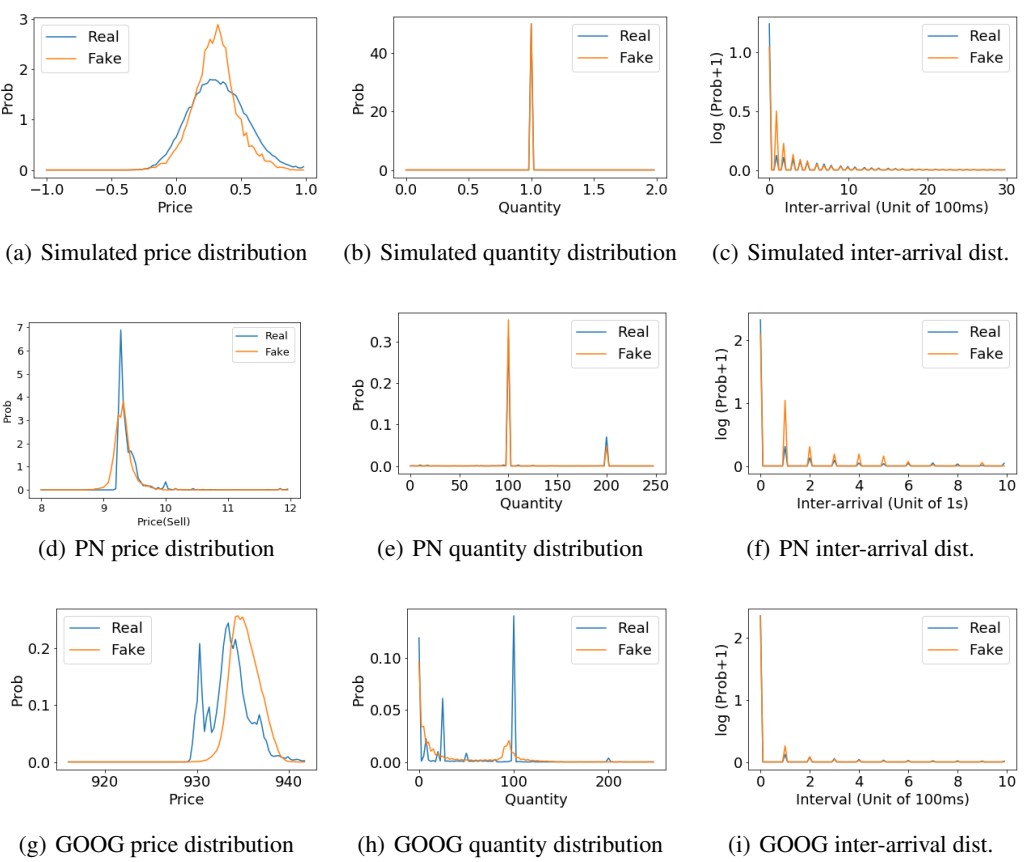

Figure 4: Simulated, PN, and GOOG submitted sell-order statistics.

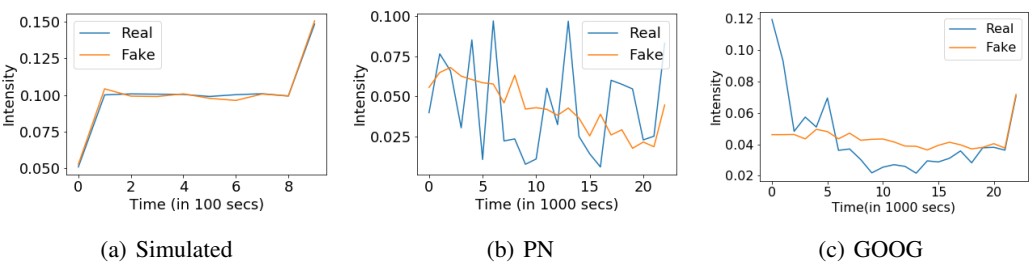

Figure 5: Intensity of market activities that include all types of orders across the trading period.

differences are particularly large for PN, likely due to the relatively smaller magnitude of trading volume for that stock.

In Figure 6, we show the change in best buy/ask as a function of time for the simulated, PN, and GOOG markets. The generated results looks similar to real data in range and variation over time for simulated data. The similarity to real best bid/ask is better for GOOG than PN, which could possibly be due to more data available for GOOG.

**Quantitative measures**: The figures till now show that the price distribution appears like a normal distribution and the inter-arrival time appears like a geometric distribution (geometric is discrete version of exponential). We fit these standard distributions to the real price and inter-arrival distribution and compare the total variation (TV) distance between the real and fitted vs real and generated

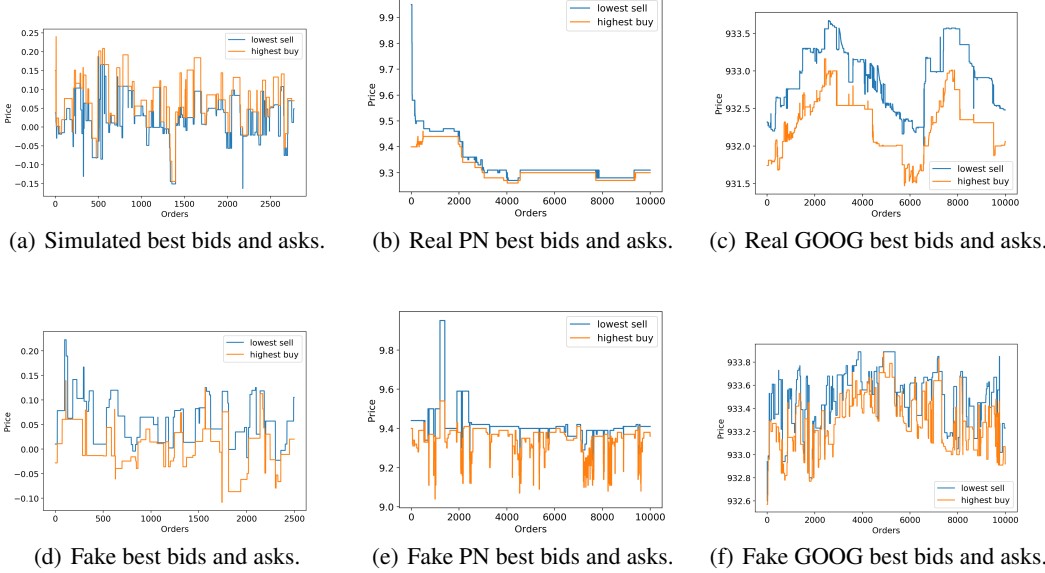

Figure 6: Best bid and ask evolution across order book state changes.

| TV distance between | Simulated | | PN | | GOOG | |
|---|---|---|---|---|---|---|
| | Price | IA | Price | IA | Price | IA |
| Real and Fitted (buy) | 0.4910 | 0.8457 | 1.3449 | 1.0571 | 1.0573 | 1.2953 |
| Real and Generated (buy) | 0.7439 | 0.2847 | 1.6828 | 0.2373 | 1.0614 | 0.3631 |
| Real and Fitted (sell) | 0.4968 | 0.8516 | 1.5453 | 0.9912 | 1.0546 | 1.3869 |
| Real and Generated (sell) | 0.8246 | 0.2025 | 1.4813 | 0.2477 | 1.1572 | 0.3286 |

Table 1: TV distance comparisons between fitted and generated distribution. IA means inter-arrival.

distributions. The quantity distribution does not appear like any standard distribution, hence we do not evaluate it by fitting. The results in Table 1 show that the generated price distribution is almost as close to the real one as the fitted price distribution. The generated inter-arrival distribution is much closer to the real one than the fitted price distribution. A point to note is that the actual price and quantity is a stochastic process with dependence on history, thus, the fitted distributions will not be helpful in generating the correct intensities or best bid and best ask evolution.

**A note on architectural choices**: Various parts of our architecture were developed iteratively to improve the results that we obtained in a previous iteration. The input of $\Delta$t to the generator and critic is critical to get the time trend in the intensity for the GOOG stock. The CDA network and the best bid and ask in history was added to improve the results for best bid/ask variation over time.

**Comparision with baseline**: We also implemented a variational recurrent generative network but found its performance to be worse than our approach (shown in Appendix B).

## 5 CONCLUSION

Our results reveal that GANs can be used to simulate a stock market. While our results are promising, there are open issues that provide for further research material. One experimental aspect is to try different size of the network in the WGAN, possibly dependent on the data size of the given stock and testing with many different variety of stocks. Another open research issue is to output cancellations in a more intelligent manner than the heuristic approach we use now. Overall, our work provides fertile ground for future research at the intersection of deep learning and finance.

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

## A ADDITIONAL RESULTS

Below we show results for buy order cancellation and sell order cancellation using the exact same measures as for the buy and sell orders in the main paper. The results also are similar to buy or sell results earlier.

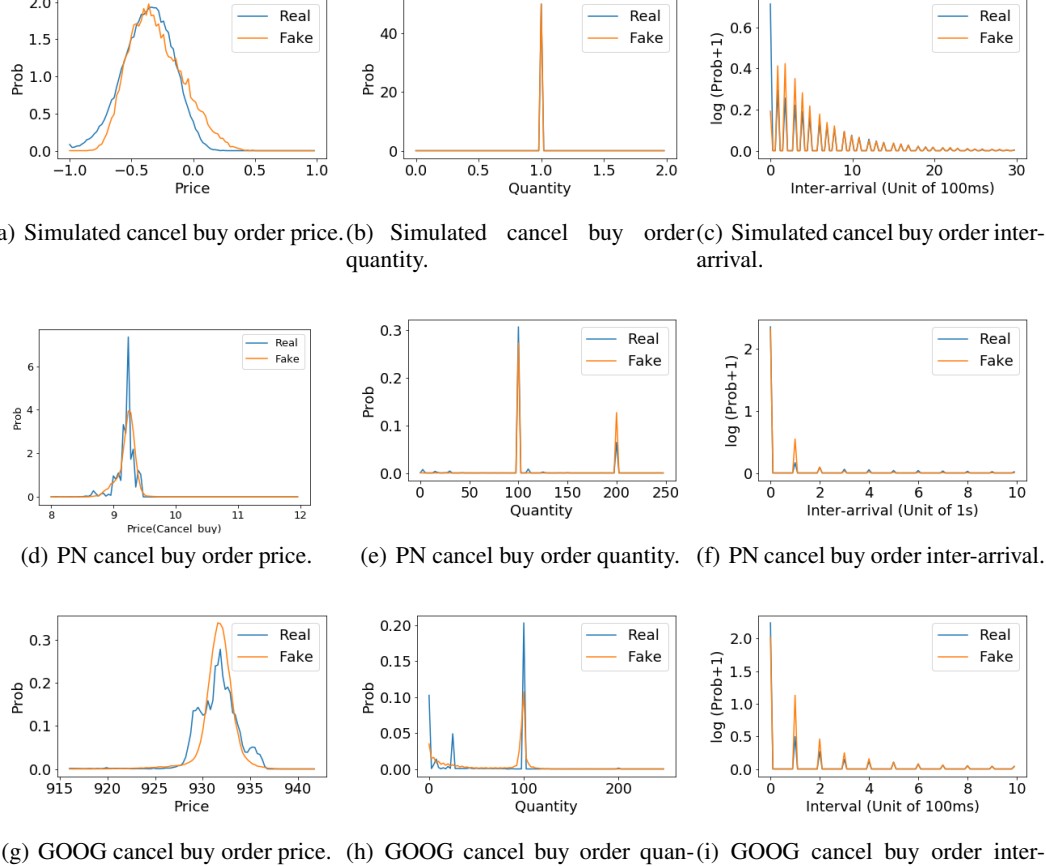

(a) Simulated cancel buy order price.
(b) Simulated cancel buy order quantity.
(c) Simulated cancel buy order inter-arrival.

(d) PN cancel buy order price.
(e) PN cancel buy order quantity.
(f) PN cancel buy order inter-arrival.

(g) GOOG cancel buy order price.
(h) GOOG cancel buy order quantity.
(i) GOOG cancel buy order inter-arrival.

Figure 7: Simulated, PN, and GOOG cancelled buy orders statistics.

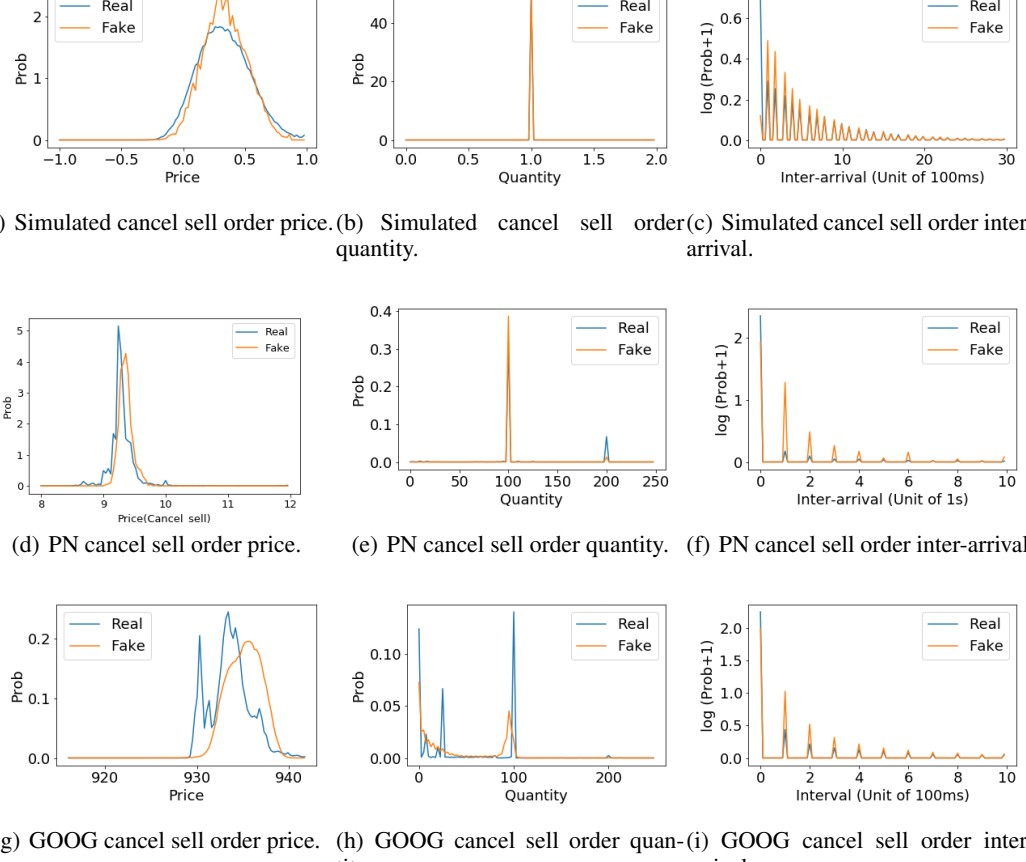

(a) Simulated cancel sell order price. (b) Simulated cancel sell order quantity. (c) Simulated cancel sell order inter-arrival.

(d) PN cancel sell order price. (e) PN cancel sell order quantity. (f) PN cancel sell order inter-arrival.

(g) GOOG cancel sell order price. (h) GOOG cancel sell order quantity. (i) GOOG cancel sell order inter-arrival.

Figure 8: Simulated, PN and GOOG sell cancel orders.

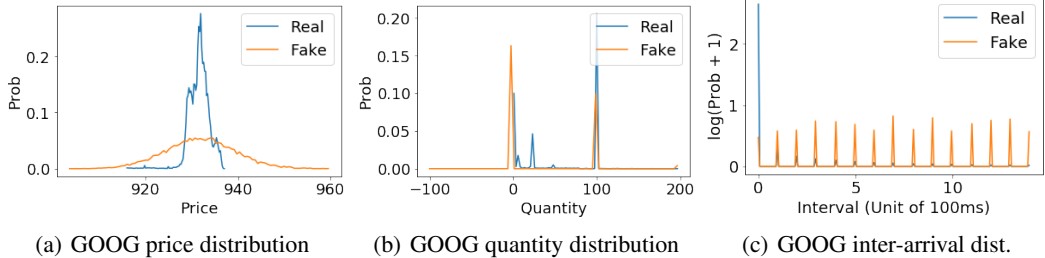

(a) GOOG price distribution  (b) GOOG quantity distribution  (c) GOOG inter-arrival dist.

Figure 9: Simulated, PN, and GOOG submitted buy-order statistics using recurrent VAE.

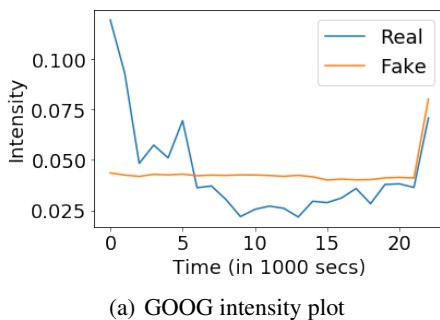

(a) GOOG intensity plot

Figure 10: Intensity of market activities for GOOG using recurrent VAE.

## B  VARIATIONAL RECURRENT NEURAL NETWORK

We use the variational recurrent network as another baseline generative model. The architecture is exactly same as the work Chung et al. (2015). We used the code available at https://github.com/phreeza/tensorflow-vrnn, but modified it. Our modification was to enable not forcing the output to be Gaussian as done in Chung et al. (2015), as those produced much worse results. Instead, we use a MSE loss. We also modified the input size, etc. to make the neural network structure compatible with our problem. The exact change to the code changing the loss function is shown below:

```
kl_loss = tf_kl_gaussgauss(enc_mu, enc_sigma, prior_mu, prior_sigma)
# we replace the maximum likelihood loss with the mse loss below
mse_loss = tf.losses.mean_squared_error(y,dec_rho)
return tf.reduce_mean(kl_loss + mse_loss)
```

The results in Figure 9 for GOOG buy order only and in Figure 10 for all types of GOOG orders shows that the entropy of the output is high (when comparing price and inter-arrival distributions) and the performance is worse than our GAN. In particular, the generated (fake) price distribution is wider than the real one (or the one generated by the GAN). The generated inter-arrival distribution is almost uniform over the discrete time points and not concentrated at 0. The quantity distribution matches the real one, somewhat similarly like our GAN approach, but it generates some negative values unlike our GAN approach (which could be discarded). The intensity distribution is also somewhat close to the real intensity. The results are similar for other types of orders.

## C    CODE SNIPPETS

Here we present codes snippets that show the architecture of the GAN. First, we start with the CDA network that is trained independently with MSE loss:

```python
input_his = Input(shape=(8,))

G = Sequential(name='discriminator')
G.add(Dense(256*3,input_dim=8))
G.add(BatchNormalization())
G.add(Activation('relu'))
G.add(Reshape((16, 16, 3)))
G.add(Conv2D(128,(3,3),padding='same'))
G.add(BatchNormalization())
G.add(Activation('relu'))
G.add(Conv2D(64, (3,3),padding='same'))
G.add(BatchNormalization())
G.add(Activation('relu'))
G.add(Conv2D(32,(3,3),padding='same'))
G.add(BatchNormalization())
G.add(Activation('relu'))
G.add(Flatten())
G.add(Dense(4))
output_vec = G(input_his)

self.net = Model(inputs=input_his, outputs=output_vec)
optimizer = Adam(0.0001)
self.net.compile(optimizer=optimizer, loss='mean_squared_error')
self.net.summary()
```

Input LSTM structure for both Generator and Critic are shown below

```python
########### Input for both Generator and Critic #######################
# history orders of shape (self.historyLength, self.orderLength)
history = Input(shape=(self.historyLength, self.orderLength), \
    name='history_full')
# current time slot: Integer, from 0 to 23
history_input = Input(shape=(1,), name='history_time')
# noise input of shape (self.noiseLength)
noise_input_1 = Input(shape=(self.noiseLength,), name='noise_input_1')

# Real order of shape((self.mini_batch_size,self.orderLength)
truth_input = Input(shape=(self.mini_batch_size,\
    self.orderLength,1),name='truth_input')

# lstm at Generator to extract history orders features
lstm_output = LSTM(self.lstm_out_length)(history)

# lstm at Critic to extract history orders features
lstm_output_h = LSTM(self.lstm_out_length,name='lstm_critic')(history)

# concatenate history features with noise
gen_input = Concatenate(axis=-1)([history_input,lstm_output,noise_input_1])
```

The Generator structure is shown below, which includes the trained CDA network

```python
############# Generator #######################
# Input: gen_input, shape(self.noiseLength+self.lstm_out_length + 1)
```

```python
# Output: gen_output_1, shape(self.mini_batch_size,self.orderLength - 4)
dropout = 0.5
G_1 = Sequential(name='generator_1')
G_1.add(Dense((self.orderLength-4)*self.mini_batch_size*100, \
    input_dim=self.noiseLength+self.lstm_out_length + 1))
G_1.add(BatchNormalization())
G_1.add(Activation('relu'))
G_1.add(Reshape((int(self.mini_batch_size), int(self.orderLength - 4), 100)))
G_1.add(UpSampling2D())
G_1.add(Dropout(dropout))
G_1.add(UpSampling2D())
G_1.add(Conv2DTranspose(32, 32, padding='same'))
G_1.add(BatchNormalization())
G_1.add(Activation('relu'))
G_1.add(Conv2DTranspose(16,32 , padding='same'))
G_1.add(BatchNormalization())
G_1.add(Activation('relu'))
G_1.add(Conv2DTranspose(8, 32, padding='same'))
G_1.add(BatchNormalization())
G_1.add(Activation('relu'))
G_1.add(MaxPooling2D((2,2)))
G_1.add(Conv2DTranspose(1, 32, padding='same'))
G_1.add(Activation('tanh'))
G_1.add(MaxPooling2D((2,2)))

gen_output_1 = G_1(gen_input)

#CDA network(train offline)
#Input: cda_input, shape(self.mini_batch_size, 8)
#Output: gen_output_2, shape(self.mini_batch_size, 4)
G_2 = Sequential(name='orderbook_gen')
G_2.add(Dense(256*3,input_dim=8))
G_2.add(BatchNormalization())
G_2.add(Activation('relu'))
G_2.add(Reshape((16, 16, 3)))
G_2.add(Conv2D(128,(3,3),padding='same'))
G_2.add(BatchNormalization())
G_2.add(Activation('relu'))
G_2.add(Conv2D(64, (3,3),padding='same'))
G_2.add(BatchNormalization())
G_2.add(Activation('relu'))
G_2.add(Conv2D(32,(3,3),padding='same'))
G_2.add(BatchNormalization())
G_2.add(Activation('relu'))
G_2.add(Flatten())
G_2.add(Dense(4))

# extract the last best bid/ask from history as the history of CDA
orderbook_history = Lambda(lambda x: x[:,-1,5:], output_shape=(4,))(history)
# gen_output_1 is output of generator
gen_output_reshaped = Reshape((self.orderLength-4,))(gen_output_1)
# remove time as it is not needed for CDA network
gen_output_without_time = \
    Lambda(lambda x: x[:,1:], output_shape=(4,))(gen_output_reshaped)
cda_input = Concatenate(axis=1)([gen_output_without_time,orderbook_history])
gen_output_2 = G_2(cda_input)

#Output of Generator, shape(self.mini_batch_size, self.orderLength) concatentated
# with output of the CDA network to get final output
```

```
gen_output = Concatenate(axis=2)([gen_output_1,\
    Reshape((self.mini_batch_size, 4, 1))(generator_output_2)])
```

The structure of the critic is shown below

```
############## Critic ##################
# Input of Critic, merge history_input, lstm_output_h and gen_output/truth_input
discriminator_input_fake = (Concatenate(axis=2)\
    ([Reshape((1, 1,1))(history_input), \
    Reshape((1, self.lstm_out_length,1))(lstm_output_h), gen_output]))
discriminator_input_truth = Concatenate(axis=2)\
    ([Reshape((1, 1,1))(history_input), \
    Reshape((1, self.lstm_out_length,1))(lstm_output_h), truth_input])
#random-weighted average of real and generated samples - following
# Improved WGAN work
averaged_samples = RandomWeightedAverage()\
    ([discriminator_input_fake, discriminator_input_truth])

#Critic
#Input: discriminator_input_fake/discriminator_input_truth
#Ouput: score
D = Sequential(name='discriminator')
D.add(Conv2D(512,(3,3),padding='same',  input_shape=(self.mini_batch_size, \
        self.orderLength+self.lstm_out_length+1,1)))
D.add(Activation('relu'))
D.add(Conv2D(256, (3,3),padding='same'))
D.add(Activation('relu'))
D.add(Conv2D(128,(3,3),padding='same'))
D.add(Activation('relu'))
D.add(Flatten())
D.add(Dense(1))
#self.D = D

discriminator_output_fake = D(discriminator_input_fake)
discriminator_output_truth = D(discriminator_input_truth)
averaged_samples_output = D(averaged_samples)

#Def gradient penalty loss
partial_gp_loss = partial(self.gradient_penalty_loss,
                          averaged_samples=averaged_samples,
                          gradient_penalty_weight=1)
partial_gp_loss.__name__ = 'gradient_penalty'
```

The full model

```
############### Model Definition  ###############
# Generator model
# Input: [history_input,history,noise_input_1]
# Output: gen_output
self.gen = Model(inputs=[history_input,history,noise_input_1], outputs= gen_output)
#Model Truth:
self.model_truth = Model(inputs=[history_input,history,noise_input_1,truth_input],\
    outputs=[discriminator_output_fake,discriminator_output_truth,\
    averaged_samples_output])
#Model Fake:
self.model_fake = Model(inputs=[history_input,history,noise_input_1],\
    outputs= discriminator_output_fake)
#Optimizer
optimizer = Adam(0.0001, beta_1=0.5, beta_2=0.9)
```

```python
#Compile Models
#Generator
self.gen.compile(optimizer=optimizer, loss='binary_crossentropy')
self.gen.summary()
#Model Truth - Generator is not trainable here
for layer in self.model_truth.layers:
    layer.trainable = False
self.model_truth.get_layer(name='discriminator').trainable = True
self.model_truth.get_layer(name='lstm_critic').trainable = True
self.model_truth.compile(optimizer=optimizer, \
    loss=[self.w_loss,self.w_loss,partial_gp_loss])
#Model Fake - critic is not trainable here
for layer in self.model_fake.layers:
    layer.trainable = True
self.model_fake.get_layer(name='discriminator').trainable = False
self.model_fake.get_layer(name='lstm_critic').trainable = False
self.model_fake.compile(optimizer=optimizer, loss=self.w_loss)
#print summary
self.model_fake.summary()
self.model_truth.summary()
```

