# OpenReview forum: "Generating Realistic Stock Market Order Streams"
_ICLR.cc/2019/Conference_

### Official Review · AnonReviewer3 · 2018-10-14
**Authors did not care about reproducibility and comparison to baselines**

**Rating:** 4
**Confidence:** 5

**Review:**

The problem addressed in this paper is worth the attention of the community. Not so much for it being of strong interest to the majority of ICLR attendees, but due to the fact that it deals with data of origin (finance) and properties (high-order Markov chain dependencies) that have never been considered in the past.

However, apart from this merit, the paper as is stands now suffers from major prohibitive shortcomings. Specifically, the authors have failed to provide a detailed account of the novel network architecture introduced in this paper. Their description is too vague, and misses crucial details. For instance, they use a convolutional structure (layer?) at some point. How is this configured? Why do they need a convolutional layer, since they present it with a vector output from a preceding dense layer? What about the CDA network (both its configuration and its training procedure)? These are crucial aspects that the authors have failed to describe at all.

In the same vein, technical details concerning the selection of the training algorithm hyper-parameters are also missing from the experimental section. Although not as glaring an omission as the previously described ones, these are also crucial for the replicability of the presented research results.

Finally, the authors have failed to provide comparisons to alternative baselines. For instance, why not train a simple LSTM and use to generate new samples. Why not use a recurrent variational autoencoder? Eventually, since the time-series we are dealing with are governed by a high-order Markov chain, we not fit and sample from a high-order hidden Markov model? These are crucial questions that must be adequately addressed by the authors.

---

> ### Author Response · Authors · 2018-11-10
> **Response to Review**
>
> We thank the reviewer for the thoughtful comments.
>
> Our plan is to release the code for full reproducibility. If the reviewer wishes, we can update the paper with code snippets (which we feel is better than text description) of the GAN and CDA network.
>
> We also agree that another baseline like recurrent VAE would be useful comparison. We are now implementing a recurrent VAE and hope to have comparative results before Nov 23.
>
> Even without such comparison, we maintain that the paper does present meaningful progress on the question whether stock market data can be generated at all. In particular, we suggest that the answer is positive for high-volume (and hence large-data) securities such as GOOG.

---

### Official Review · AnonReviewer1 · 2018-11-02
**This is an interesting paper on the application of GAN in generating order data.  The evaluation and assumptions used in the paper need further justifications.**

**Rating:** 5
**Confidence:** 4

**Review:**

The objective of this paper is to use GAN for generating the order stream of stock market data.   The novelty of the paper is the formulation of the order stream and the use of GAN for generating the stock data.   This is a paper for the application of GAN and there are limited contribution to the technical aspect of machine learning.    The paper is clearly written.   There are two main assumptions used in the paper; one is the Markov chain and the second one is the stationary distribution.   In real case, both assumptions are unlikely  to be satisfied.  The orders are mostly affected by many external factors and financial data are known to be non-stationary.  The authors may have to justify these assumptions.

Another issue is the evaluation of the results.  The paper uses five statistics to evaluate the generated data.  What we can conclude is that the generated data follow similar statistics with the real data.   But we do not know if the generated data offer extra information about the market.  The paper has used synthetic data in the experiments.  So it means that we could have models that generate data that look like real data.  If so, what are the benefits of using GAN to generate the data ?

---

> ### Author Response · Authors · 2018-11-10
> **Response to Review**
>
> We thank the reviewer for the thoughtful comments.
>
> We agree with the reviewer that stationarity will generally not hold in the real world and the stock market is affected by external factors not represented in our data-set. However, all learning models are an approximation of the truth and in that sense our assumptions are also approximating the true world while still allowing for formal reasoning use the GAN.
>
> The statistics we used for evaluations would be considered natural measures for markets by finance specialists. Marginal distributions of price and quantity summarize the two key dimensions of an order sequence. Temporal aspects are described by inter-arrival time, the intensity change over time, and the bid-ask spread. We believe these five statistics capture a wide range of properties of the security.
>
> There are several potential benefits of a GAN-learned model for generation beyond using a simulator. One is that simulations necessarily incorporate much more modeler-imposed structure than the GAN. Others include employing the GAN for valuable tasks such as anomaly detection, or for generating synthetic data for a variety of purposes—such as calibrating a simulator.

---

### Official Review · AnonReviewer2 · 2018-11-06
**Review: GENERATING REALISTIC STOCK MARKET ORDER STREAMS**

**Rating:** 5
**Confidence:** 4

**Review:**

This paper proposes a Generative Adversarial Network (GAN) methodology to learn the distribution of limit orders that arrive on an equity market. The proposed approach captures the (mostly discrete) structure in the generated orders; modeling is carried out with a conditional Wasserstein-GAN with a recurrent neural networks in both the generator and critic used to capture the time dependency, and convolutional layers to capture other conditioning information. Experiments are provided on both synthetic and real data.

Overall, the paper is well written and easy to follow. The application of WGAN to modeling financial market microstructure is novel and appropriate: microstructure is probably one of the areas of finance where generative models can be trained with ample data, and overfitting risks can therefore be controlled to some extent. In this respect, the paper brings a valuable contribution, both to finance by proposing a direct generative model of a complex process, and to machine learning by showing a useful application of GANs outside of the traditional domains.

My main reservation with the paper is that the experimental results could be more convincing. In particular, results for real data are shown for only two stocks, representing respectively 20k and 230k orders. Although the trained model captures some aspects of the distribution (e.g. inter-arrival times), it misses others, such as important aspects of the (univariate) price distribution, as well as the best bid-ask dynamics. As they stand, the results appear mostly anecdotal, and lack either a breadth or depth of analysis. It would be sensible for the authors to:

1. Compare their models to other limit-order book simulation models proposed in the financial econometrics literature;
2. Characterize the sensitivity of the learned distribution on the architecture hyperparameters.

For these reasons, it appears difficult to recommend acceptance of the paper in its current state.

---

> ### Author Response · Authors · 2018-11-10
> **Response to Review**
>
> We thank the reviewer for the thoughtful comments.
>
> We agree that the evidence as presented appears anecdotal and that comparative results would be illuminating (see response to R3). Nevertheless, we regard the generated bid-ask spread data as reasonably faithful on an absolute basis, and the same for the price distribution for GOOG. GOOG has lot more data than PN, as the reviewer notes. As stated in the paper, the low number of data points for the small cap PN leads to results that are off and not quite unexpected from a learning perspective. Moreover, the real market data is much more noisy than the synthetic data, thus matching real data as closely as we match with synthetic data may not just be possible or even desirable for reasons of over-fitting.
>
> With regards to the simulator suggestions, the financial market simulator we are using has been well-established in agent-based financial modeling literature. Also, the simulator configuration has not been calibrated to the particular real data employed here, as is apparent from statistics shown.

---

### Author Response · Authors · 2018-11-26
**Revised version based on reviewers comments.**

In response to the reviewers comments, we have updated our submission with the following information:
1) Appendix B: presented results with a recurrent VAE, which performs worse in certain aspects compared to our approach.
2) Appendix C: posted code snippets presenting our neural network,loss functions, and optimizer parameters. We will link to the publicly available code when anonymity is not a concern.
We hope the reviewers take these improvements into account.

---

### Meta-Review · Area_Chair1 · 2018-12-20

**Confidence:** 5
**Recommendation:** Reject

**Metareview:**

The reviewers raised a number of major concerns including the incremental novelty of the proposed (WGANs are applied to a new domain), and, most importantly, insufficient and unconvincing experimental evaluation presented (including the lack of comparative studies). The authors’ rebuttal failed to fully alleviate reviewers’ concerns. Hence, I cannot suggest this paper for presentation at ICLR.